# Sustainable Curriculum Planning for Artificial Intelligence Education: A Self-determination Theory Perspective

**Thomas K.F. Chiu * and Ching-sing Chai**

Department of Curriculum and Instruction, Faculty of Education, Chinese University of Hong Kong, Hong Kong, China; cschai@cuhk.edu.hk
* Correspondence: tchiu@cuhk.edu.hk

**Abstract:** The teaching of artificial intelligence (AI) topics in school curricula is an important global strategic initiative in educating the next generation. As AI technologies are new to K-12 schools, there is a lack of studies that inform schools' teachers about AI curriculum design. How to prepare and engage teachers, and which approaches are suitable for planning the curriculum for sustainable development, are unclear. Therefore, this case study aimed to explore the views of teachers with and without AI teaching experience on key considerations for the preparation, implementation and continuous refinement of a formal AI curriculum for K-12 schools. It drew on the self-determination theory (SDT) and four basic curriculum planning approaches—content, product, process and praxis—as theoretical frameworks to explain the research problems and findings. We conducted semi-structured interviews with 24 teachers—twelve with and twelve without experience in teaching AI—and used thematic analysis to analyze the interview data. Our findings revealed that genuine curriculum creation should encompass all four forms of curriculum design approach that are coordinated by teachers' self-determination to be orchestrators of student learning experiences. This study also proposed a curriculum development cycle for teachers and curriculum officers.

**Keywords:** artificial intelligence education; curriculum planning; curriculum design; teacher education; self-determination theory; teacher belief; K-12 education

## 1. Introduction

Artificial intelligence (AI) has emerged as a ubiquitous form of technology in our everyday lives. Many educators and education authorities have begun considering including AI topics in K-12 curricula to prepare school students to learn about these emerging technologies. Such initiatives inevitably involve curriculum planning. As AI is an emerging field undergoing rapid changes, and considering that teachers are most likely not familiar with its content, understanding how existing theoretical frameworks of curriculum planning and teachers' perspective can be invoked to respond to the situation would be of interest to the refinement of curriculum theories and teacher development. While innovators among teachers are creating AI curricula, a recent review of AI in education has highlighted the lack of study on teachers' perspective [1]. It is important for researchers to document the teachers' perspective, as it undergirds their sense-making of the emerging AI technology for curriculum planning [2]. This study thus attempts to understand the teachers' choice of action in planning AI curricula and the personal reasoning behind the teachers' effort.

On the sociopolitical front, both China and the United States announced relevant AI education initiatives in 2018. The Ministry of Education of China announced the "Artificial Intelligence Innovation Action Plan for Institutions of Higher Education" to encourage and support young people to participate

in AI work, and school teachers to teach their students AI knowledge. In response, the Association for the Advancement of Artificial Intelligence (AAAI) and the Computer Science Teachers Association (CSTA) formed a joint working group to develop national guidelines for the teaching of AI to K-12 students. These projects aimed to contribute to the development of AI-related school curricula. Including AI topics in school curricula is an important global strategic initiative in educating the next generation [3]. AI education in schools not only helps children understand what the AI technologies are and how they work, but it also inspires future AI researchers, ethical designers and software developers [3]. However, curriculum design for K-12 schools is more complex compared to higher education. It involves considerations of how the new initiative or policy translates into practice, and considerable variation in implementation can be expected from school to school [4]. Schools often have fixed and inflexible timetables and subjects, and limited resources with regard to classroom equipment. In addition, while AI is an established field in higher education, school teachers are not formally trained for AI education. Integrating technology is still currently viewed as problematic and it is important to understand teachers' value-driven and feasibility assessment processes embedded within dynamically evolving school environments [2]. Building on their work, integrating AI has unique challenges in that it is totally new to schools, with the AI content not defined and the teachers having to figure out where it fits in a crowded curriculum. Therefore, designing AI-related school curricula is very challenging for school teachers, school leaders, education officers, policy-makers and AI experts, and it is important to raise the challenges teachers face to facilitate curricula planning work.

Most recent studies related to AI curricula focused on what content knowledge and skills should be included [5] and what AI tools are more effective for student learning [6]. These studies viewed teaching as transmission of knowledge and used the syllabus and assessment methods to plan their curriculum through identifying appropriate content and effective delivery methods, and enhancing students' competencies. They focused on predefined content and outcomes, rather than how teachers, students and knowledge interact [7]. In other words, the current approach to AI curriculum planning may neglect teachers' perspective and sense-making, and also students' agency in their learning [7,8]. Accordingly, these recent AI curriculum studies do not inform us well about the overall design of a formal curriculum and its planning approach for this emerging subject. Besides, school curriculum planning is fundamentally a political process [4], one which involves arguments about questions of value. Therefore, teachers' beliefs and views will decide what the curriculum looks like [9]. Teachers' intrinsic motivation is critical in the planning of curricula for sustainability [10] because AI curriculum design requires an iterative development cycle. This motivational process can be explained by the self-determination theory (SDT), which provides a theoretical framework to explain the teachers' fundamental psychological needs—autonomy, relatedness and competence—for educational innovation [11,12].

As such, we used a qualitative study to explore the views of teachers with (AI teachers) and without (non-AI Teachers) AI teaching experience on key considerations for the design, implementation and revision of a formal AI curriculum for K-12 schools. AI teachers' views were sought on developing, implementing and redeveloping the school-based curriculum, and non-AI and AI teachers' views were sought on teachers' preparation issues (e.g., teacher feelings and perceived needs). In others words, this study is concerned with (1) curriculum preparation—how to prepare and motivate teachers to design and teach the curriculum, (2) curriculum development—what to include in content knowledge and what effective learning designs should be adopted and (3) curriculum renewal—how to motivate teachers and what teachers need to renew the curriculum. Situating the case on the nexus of theory and practice, we attempt to use the four basic curriculum planning approaches and the self-determination theory (SDT) to explain the research problem and findings in this paper.

## 2. Theoretical Frameworks

### 2.1. Self-determination Theory (SDT)

To understand how to prepare and motivate teachers to iteratively develop and implement AI curricula, we used SDT, which provides a theoretical framework of motivation to explain their psychological needs for educational innovation [11,12]. SDT posits that all individuals possess three fundamental psychological needs—autonomy, relatedness and competence. These needs determine individuals' motivations to act or not act [11]. Autonomy refers to a sense that individuals can control and exercise their freedom of choice to proceed in whatever way they see as best. Relatedness refers to a sense that individuals are connected, belong to a community and share the community's purpose in their efforts. Competence refers to a sense that individuals have the knowledge and skills necessary to successfully accomplish a task. When all the three psychological needs are met, teachers are intrinsically motivated to sustain their own personal growth and well-being, which may enhance the development of curricula. Accordingly, schools can foster teachers' intrinsic motivation to teach and develop the curriculum by supporting their psychological needs for autonomy, competence and relatedness.

The implications of SDT for introducing innovations advocate school leaders to use autonomy-supportive, rather than controlling, strategies [10]. In an autonomy-supportive environment, school leaders should consider teachers' perspectives, allow for choices around planning and reduce unnecessary stress and demands on teachers. Teachers should make their own choices and decisions with regard to the curriculum based on their self-efficacy and intended development, and in return feel empowered in planning [9,13]. Moreover, schools can further support competence by providing teachers with the necessary professional training and freedom [9,10]. For example, schools can allow and support teachers to take the courses they want to join during school days. Relatedness, the last psychological need, has received less explicit focus in the literature, often being discussed in terms of mentorship, group experience and collaborative learning [14]. In these studies, group experiences are seen as having a self-oriented purpose linked to consideration of personal benefits. Accordingly, taking other perspectives into consideration in relatedness is understudied; current research on SDT and education tends to consider a more self-oriented purpose (getting from the community) rather than an other-oriented purpose (giving to the community).

Overall, teachers who are empowered to internalize their experiences in curriculum planning activities are more likely to sustain the development of the curriculum. Other than understanding teachers' motivation to participate in this educational innovation, the other factor is curriculum planning approaches.

### 2.2. Four Curriculum Planning Approaches

"Curriculum" refers to all experiences that are planned and guided by a teacher and learned by students, whether in a group or individual setting, or in inside or outside classrooms [7]. Curriculum theory, derived from educational, philosophical, psychological and sociological perspectives, is fundamentally concerned with values and ways of viewing educational curricula and policy decisions. Literature outlines four approaches to understanding curricula: curriculum as content, product, process and praxis. These approaches are used independently or in an integrated manner to theorize curricula development in schools [7,15]. Regardless of the approaches, understanding curriculum development involves unpacking the underlying relationships between purposes, knowledge and pedagogy.

The curriculum as content approach sees education as transmission of knowledge. Curriculum planning is thus the construction of a syllabus (a body of subject content) and the identification of effective delivery methods [7,15]. Its supporters are more likely to follow a textbook approach of an order of contents, or a knowledge structure approach to a subject. They tend to limit their planning to the consideration of the body of knowledge that they want to deliver. The justification for the curriculum lies in its content but not its effects. This view of curricula is very popular amongst teachers in primary (Grade 1–6) schools [7].

The curriculum as product approach sees education as instrumental to enhancing students' competencies. It focuses on assessing student learning outcomes [7,15]. The curriculum is viewed as a design for a technical exercise, and it takes the performance and competence of students as the core components [16]. It aims to prepare students adequately for specific activities, and it involves detailed attention to what the students need to learn and know in order to pursue further study, work, live their lives and so on. This approach is often found in many technical, skill-based and training programs where specific tasks or jobs have been identified, as well as technology and engineering subjects where the body of knowledge and concepts are well defined. The curriculum often draws up lists of competencies and informs students what they must learn and how they will do it; therefore, the students have little or no voice in their learning. By having predefined outcomes, this approach tends to direct attention to teaching.

The two approaches discussed earlier usually generate a set of documents for implementation. John Dewey's progressive and student-centered approaches, on the other hand, spurred the curriculum as process approach [7]. This approach sees education as development and it focuses on how teachers, students and knowledge interact, rather than on delivery of predefined content and outcomes. Learning objectives have a tendency to change as the triadic relationships evolve [7]. The curriculum is not a standard package of materials that needs to be consistently covered and delivered in classrooms, but a specification about teaching practice [15]. It is seen in terms of what actually happens in classrooms and what teachers and their students do to prepare and evaluate subject matter. For example, choices of content depend on what fits student needs and interests; learning outcomes are developed from a collaboration between teachers and students, but not applied to all the students. In this approach, students are not treated as objects but as subjects who have voices. This approach shifts the focus of the curriculum from teaching to learning.

The process approach emphasizes interpretation and meaning-making, and does not make clear statements about the interests it serves. Bringing this issue to the center of the process, the curriculum as praxis approach sees education as committed action and focuses on making sense of the knowledge in the learning process by connecting it to real-world applications [15]. Under this approach, students and teachers reflect together and develop the problem-solving strategies and skills that they use to solve real-world problems. They are required to work out an action plan for acquiring the content knowledge and achieving the outcomes. The learning process and outcomes are continually evaluated.

Adopting a particular curriculum planning approach has a major influence on pedagogy [4]. For example, the content approach encourages teacher-centered approaches to teaching, the product approach places heavy emphasis on drills and practice, the process approach leads to the design of student-centered learning activities and the practice approach tends to adopt problem-based learning. However, these four approaches to curriculum planning are not mutually exclusive [7,15]. For example, supporters of the process approach would not argue that content and assessment are unnecessary and negligible, but the selection of content is a secondary consideration. The first two approaches adopt a behavioral stance and structured teaching, and set objectives and attainment targets that must be taught to students. The last two approaches draw on student-centered learning theory and educational and developmental psychology. They identified and nurtured the strengths of students, with every student taking an active role in her/his learning and with both students and teachers developing the curriculum.

Contemporary education favors the process and praxis approaches over the content and product approaches, which is evidenced by a massive shift from the content and praxis approaches to the process and product approaches in teaching practices and educational reforms all over the world [8]. However, school curriculum planning, unlike higher education, is fundamentally a political process [4]. Different teachers have different views about what should be covered in the curriculum and how it should be implemented. In these approaches, the curriculum is developed based on a stable set of knowledge, such as language and science. While these existing approaches are likely to be manifested in the process of AI curriculum planning, the epistemic essence of AI technology may demand new categories of

consideration that could alter current curriculum theory. For instance, the subject matter of AI is highly dynamic and it is also highly unfamiliar to K-12 teachers. In addition, there are many ethical concerns about this form of technology, such as unemployment, AI bias, singularity and superintelligence [1,17]. Will AI replace human workers? Will AIs' decision-making be transparent? Should AI systems be allowed to kill? Will AIs evolve to surpass human beings? Therefore, how teachers approach curriculum planning for subject matter such as AI is less likely to have been accounted for.

Overall, the content and product approaches to planning tend to be adopted in technical subjects, such as physical education, and at the program level due to their focus on the competencies and assessment [16]. Moreover, the process and praxis approaches are more likely to be adopted in planning more established subjects, such as language and science, and at the classroom level because the teachers can choose the units that they want to focus on for teaching [7,15]. In other words, the teachers can decide what, how and when to teach, how to connect to students and how long to spend teaching them. However, how to plan a curriculum for any emerging subject domain or disruptive education innovation is less clear, but it is a required competency as more and more subject areas are being renewed and re-represented with technological advancements.

### 2.3. AI Teaching and Learning in Schools

To date, very few studies on AI school teaching have been conducted [1,5,6]. There are some important but fragmented findings in the research into AI curriculum planning and development. SenseTime collaborated with East China Normal University to establish AI laboratories and produced the first textbook series for high schools—*Fundamentals of Artificial Intelligence* [5]. The textbook series suggested that they were designed to prepare high school students who want to pursue further study or work in AI disciplines. The textbooks focused on technical content and skills, including advanced complex mathematics. Overall, they adopted the content and product approaches. They are inappropriate for general education, i.e., all high school students. The Massachusetts Institute of Technology [6] examined the impact of different AI learning activities, including robots, on children's learning, and they adopted more process and praxis approaches. Therefore, there is neither an existing established curriculum nor well-defined AI content knowledge for high schools. Research on the curriculum development approaches adopted, the curriculum development processes and the consequences are necessary for educators to enhance the process of integrating AI topics into K-12 education.

Moreover, AI technologies, different from other new technologies, are emerging and potentially disruptive. As machine learning becomes more powerful and narrow AI are performing more jobs, AI technologies and products are replacing jobs such as cashiers and proofreaders [17]. AI future development would cause further loss of jobs due to automation and computerization, and people need to improve AI skills to change careers. This anxiety could cause facilitating or debilitating effects [10,18]. When students perceive the AI learning as rewarding and hold a positive attitude toward AI technologies, the facilitating effect occurs; otherwise, the debilitating effect occurs [18]. The effect is dependent on the curriculum designed. This points to the importance of investigating how teachers are conceptualizing the AI curriculum and the underlying curriculum approaches that they are adopting in conjunction with the personal psychology that is driving them as teachers.

In sum, incorporating these two theoretical frameworks into the development of the educational innovation of an AI curriculum facilitates an understanding of the process and mechanisms by which innovations might work.

### 3. This Study

As we discussed earlier, AI K-12 education is new to academia and schools. There is a serious lack of relevant studies, particularly in planning, implementing and renewing AI curricula. The unique thing about AI is that it is new, emerging, interfering and disruptive, and it is definitely not part of teacher education. It thus offers an opportunity to enrich theories of curriculum design and planning

processes for subject matter that teachers do not have much prior knowledge on. Teachers' perspectives are very essential to make sense of the emerging AI technology for curriculum planning [2]. Therefore, the goal of this qualitative study was to explore the views of technology teachers with (AI teachers) and without (non-AI Teachers) AI teaching experience on key considerations for the iterative creation of a formal AI curriculum for high schools (Grade 7–12). Accordingly, the two research questions are from the teachers' perspectives.

RQ1: How do the three psychological needs—autonomy, competence and relatedness—in SDT relate to curriculum development?

RQ2: How do curriculum planning approaches relate to curriculum development?

To achieve the goal of our study, we conducted semi-structured interviews with the teachers to capture their experiences and views of planning and teaching AI topics.

## 4. Method

### 4.1. Participants and Data Collection

School curriculum planning involves arguments about questions of value [4]. Different teachers have different views about what should be covered in the curriculum for the needs of their schools and students. Therefore, we used purposeful sampling [19] to make sure that participants with all the conditions were recruited (broad categorization of the academic ability of students and AI teaching experience) from the pool of 48 partnership schools. In particular, four technology teachers with experience in AI-related teaching and four without from each banding in the pool were randomly selected (remark: secondary, i.e., Grade 7–12, schools in Hong Kong are categorized into three bandings based on student academic achievement). This resulted in 18 male and six female participants. The sample size of 24 has been recommended by Ando, Cousins and Young [20] as sufficient to generate codes for thematic analysis. The teacher participants had a minimum of four years' teaching experience, and they hold qualifications to teach information communication technology (ICT). All the participants were informed of their rights and gave agreed consent. This study had got ethical approval from the Human Research Ethics Committee at the University of Hong Kong (project identification code: E41708017). There is no conflict of interest between the author and participants. If the participants develop any concern later on, they have the right to withdraw from the study at any time.

We collaborated with two experts who had extensive experience in ICT teaching and curriculum planning to use the proposed theoretical framework—SDT and curriculum planning approaches—to develop a semi-structured interview protocol. The first expert was a senior government curriculum officer with a master's degree who believed that including AI in school curricula is necessary; the second expert, who was on a subject panel for technology in a school, believed that including AI in school curricula is necessary but would create a high workload. This protocol (see Appendix A) aimed to facilitate open discussions and to collect in-depth perspectives. The interviews explored:

1.　How to prepare new teachers to design and teach AI curricula (see the three needs in SDT).
2.　How to plan and develop the AI curriculum and its content (content and product approaches).
3.　Logistical issues within a school environment (e.g., timetable and facility activities) (see relatedness in SDT).
4.　Teaching strategies and learning design (process and praxis approaches).
5.　How to refine the curriculum in an iterative manner (see the four approaches).

To collect the data, a trained interviewer conducted 24 semi-structured interviews that were allowed to evolve as a dialogue between the participants and the researcher within the framework of the topics. In other words, the participants and the researcher jointly explored how to plan the AI curriculum (mean duration: 50 min).

### 4.2. Data Analysis

We transcribed and translated the interview data to English and used a hybrid inductive and deductive thematic analysis to identify themes related to our theoretical framework. This offers a useful method for working within a participatory research paradigm to inform policy/curriculum development [21]. Accordingly, we adopted thematic analysis using four phases, guided by the theoretical constructs, to analyze the data.

- Phase 1: becoming familiar with the data and generating initial codes. A team member (the first expert) read, re-read line by line and annotated transcripts with codes that described notable content.
- Phase 2: searching for themes. A different team member (the second expert) reviewed all annotated transcripts to thoroughly examine codes and to identify any differences in interpretations. Another team member (the first author) acted as the mediator of any differences in interpretation. The team analyzed the codes to generate initial themes.
- Phase 3: reviewing themes. The team may group some existing themes together or split some themes into subthemes. This process was repeated until the researchers were satisfied with the thematic map.
- Phase 4: defining and naming themes. The team defined and gave names that provided a full sense of the theme and its importance.

## 5. Result, Discussion and Conclusions

The final thematic map devised in the results identifies two main themes and six subthemes (see Figure 1). They were: Theme 1 (contextual factors—perceived needs, multiple professional development activities and multilevel engagement (RQ1, SDT)) and Theme 2 (curriculum design—content input and product output, process and praxis as the pathway towards meaningful learning, and renewal for betterment (RQ2, curriculum planning)). Hence, we provide six empirical implications, two theoretical contributions and one practical recommendation.

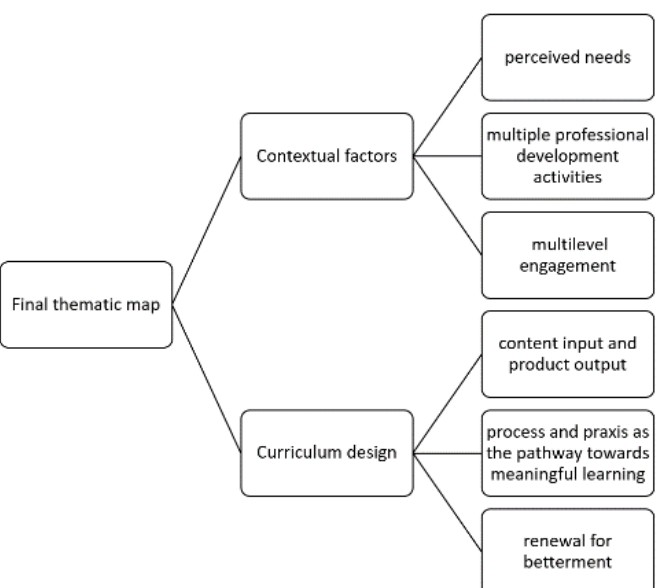

**Figure 1.** The final thematic map.

*5.1. Empirical Implications*

5.1.1. Theme 1: Contextual Factors

Theme 1 describes the various forms of needs and perceptions that the teachers are experiencing as they are confronted with the emerging AI technologies and the curriculum development work within and beyond the school context.

Theme 1.1: Perceived Needs

In this subtheme, an AI curriculum is needed for preparing school students to learn about the emerging technologies. The analysis showed that all the technology teachers expressed that students should learn more about AI due to its large impact on our everyday life, and there is a need for an AI curriculum. This is aligned with the important global strategic initiative in educating the next generation by including AI topics in school curricula [3] and the students' motivation to learn AI technologies [22,23]. This subtheme is illustrated in the below excerpts. The teachers felt teaching AI was their responsibility and endorsed this initiative. This personal endorsement or sense of choice reflects the need for autonomy in SDT [13]. This need support represents an interpersonal environment in which the teachers take students' perspectives into consideration, provide relevant information and opportunities for choice, and encourage them to accept personal responsibility [10,13].

- "Our smart phones use AI technologies, they are everywhere. Our students must master the technologies" (non-AI teacher 1).
- "Yes, I definitely think there would be a demand, . . . AI knowledge is very important for our students' careers and lives. They need to learn more about AI. I am interested to know more" (non-AI teacher 3).
- "Our students need to learn the AI technologies for their future. It is one of our jobs in schools" (AI teacher 6).
- "It is our responsibility to teach students the technologies for their future" (AI teacher 8).

Theme 1.2: Multiple Forms of Professional Development Activities

In this subtheme, the teachers perceived a strong need to earn AI knowledge and curriculum design capacity from different experts, including AI professionals, mathematics and engineering professors, and school teachers in order to provide AI teaching. The analysis revealed that both groups of teachers were qualified to teach ICT, but they felt anxious and less confident to teach their students this rapidly evolving knowledge and unanimously lacked the relevant content knowledge and curriculum design capacity, which is reflected in the following excerpts.

- "I need to receive professional development programs to learn more about AI before I design and teach the topic . . . . I don't think I am capable of teaching this topic" (non-AI teacher 5).
- "The concepts of AI are unclear for me. I would like to know more. Sometimes, I was not sure if I explain the (AI) knowledge to my student well in the classrooms" (non-AI teacher 7).
- "There are so many different tools for learning AI. I would like to receive the relevant workshops. I did not feel comfortable to design and teach my lessons with the tools... I wanted to know more about how to design an appropriate curriculum" (AI teacher 6).
- "I am not confident to teach AI topics. . . . I need to learn more about AI and their applications" (AI teacher 7).

This is aligned with the findings of [10] that teachers' efficacy and anxiety are the key factors that need to be addressed in promoting educational innovation. How the teachers felt can be explained by competence in SDT. AI knowledge is emerging and interdisciplinary, as is evident for example in the definition of AI and machine learning, and different new AI tools; therefore, the technology teachers perceived that they were neither qualified to teach AI nor able to design appropriate tasks and curricula (see competence in SDT). A plausible explanation is teacher anxiety about the emerging

technologies that is strongly associated with their perceived efficacy in teaching [9,10]. The emerging technologies may be perceived as posing challenges to teaching and confusing students because young children require a stable foundation of knowledge to build new technical knowledge. The teachers expressed that they needed to learn from different experts (see the following excerpts).

- "I believe I need to learn more about AI from other school teachers before I design and teach the topic" (non-AI teacher 2).
- "I hope to attend more workshops run by XXX (AI industry)" (AI teacher 2).
- "University professors could give us a talk" (AI teacher 3).
- "I attended a few lectures given by a mathematics professor and an AI professor. The lectures helped me a lot" (AI teacher 9).

Theme 1.3: Multilevel Engagement

Multilevel engagement is the last subtheme in Theme 1, and it indicates that engaging school leaders, other teachers, university professors and industrial partners in the development process is necessary. They also suggested that it is crucial to engage school leaders in the curriculum design process through discussions for their endorsement (see the below excepts of school leaders' endorsement) to establish and maintain support communities with AI professionals and other school teachers on AI tools and pedagogy for sharing and learning good practice (see the below excepts of peers' learning and advice from university professors and AI professionals). A cross-cutting axial in the following excerpts seems to be getting support from multilevel engagement.

School leaders' endorsement:

- "My boss should know what is going with the curriculum development" (AI teacher 4).
- "I believe that engaging my principal in the development would make the job easier" (AI teacher 6).
- "We need the support from our panel and principal to design and deliver the curriculum" (AI teacher 9).

Peers' learning:

- "I want to learn more about how to design AI curricula. Perhaps some sharing from other colleagues" (non-AI Teacher 1).
- "Definitely, I need more training on this topic and its pedagogy run by school teachers" (AI teacher 3).
- "I was a member of a Whatsapp group with other school teachers. They shared the latest AI tools and teaching ideas" (AI teacher 5).

Advice from university professors and AI professionals:

- "Professors could give us some advice" (non-AI teacher 3).
- "I hope to attend AI tools workshops run by XXX (AI industry)" (AI teacher 2).

The teachers expressed that the curriculum design process involves changes in school organizations (e.g., subject assimilation and timetable adjustment) and an increase in pedagogical and technical knowledge. There are only a few technology teachers within a school; the teachers working alone can feel very isolated. Working in a team or a community of practice with other teachers, AI experts and professionals will provide support (see relatedness in SDT), which is essential in this curriculum design process. According to relatedness in SDT, the teachers are more motivated when they feel connected and that they belong to a community. Therefore, helping the teachers engage with important relationship partners, such as school leaders and university professors, to share their abiding interests, goals, values and behaviors is necessary in the design process [12].

Moreover, the majority of the AI teachers reported that there was no room in their tight teaching schedules to teach additional AI units. The inflexible and fixed timetable and classrooms restricted their teaching spaces, and the availability of appropriate AI tools was a major issue in AI teaching (see the following excerpts).

- "It would be better to have longer lessons to teach the AI units" (AI teacher 3).
- "In my school timetable, the technology subject only has one period (around 40 min). How could I do project-based learning?" (AI teacher 8).
- "I need more funds to purchase or subscribe to the services or tools for AI teaching and learning" (AI teacher 9).
- "Some other schools have the 50 sets of robots for a class; my school only has 5 sets for my class. My students would have less hands-on experience" (AI teacher 10).

Therefore, it is very important that the teachers needed to feel connected and that they belong to the school leadership teams (see relatedness in SDT). Due to the boss and employee relationship, school leadership teams' engagement and endorsement would make the teachers feel empowered to tackle the obvious barriers, including school-based curriculum structures and logistics. This supports the psychological needs for autonomy and relatedness that would increase teachers' belief in themselves, or self-efficacy, which could encourage them to design, implement and renew the curriculum [24]. It is very important to allow teachers to make decisions in the best interest of individual schools and students based on the diverse ability levels and school environments. According to autonomy and relatedness in SDT, it is important that the teachers feel endorsed and have the freedom to identify support structures to aid them so that they feel at ease to move towards AI teaching. Sufficient and concrete logistical support from schools is necessary, and the teachers expect to receive additional resources or guidance given by schools and/or governments. Accordingly, the teachers can select the tools that are suitable for the school classroom environments and are easily assessed by their students.

5.1.2. Theme 2: Curriculum Design

Theme 2 addresses the teachers' view about curriculum planning work, including how they are trying to formulate a coherent and workable curriculum.

Theme 2.1: Content Input and Product Output

This subtheme shows that the content input and product output delimit the problem space of the curriculum design. In our analysis, the AI teachers expressed that the content and product planning was an integration pathway via the existing technology subject and the first essential step in curriculum development. It is impossible to have a new independent subject called AI in schools. AI topics should be taught in the technology subject, and the objectives and assessment of AI teaching units should be aligned with those of technology subjects, which is reflected in the following responses.

- "AI is not an independent subject, but a teaching unit under the technology key learning area" (AI teacher 1).
- "In my school, we would carefully plan and assimilate the AI teaching units into the other technology subjects" (AI teacher 7).
- "No problems with the assessment, it was as same as computer literacy" (AI teacher 9).

When the teachers began to design the AI curriculum, they used the existing technology school curriculum to identify the relevant content and assessment, and the effective instructional strategy for their students [7,15]. For example, the three main content components were knowledge in, process in, and impact of AI, which are found in the curriculum (see the following excerpts).

- "I first used the curriculum guide to identify the content and assessment" (AI teacher 1, content and product first).
- "Content development is my first task" (AI teacher 4, content and product first).
- "Students should know the background and history of the AI technologies" (AI teacher 4, knowledge in AI).
- "Students should learn about how the computer develops the ability, which includes modeling, statistics and learning algorithms. They also should learn how AI technologies process data in different aspects" (AI teacher 5, process in AI).

- "I believe my students should learn about the societal and personal impact of AI locally and globally" (AI teacher 8, impact of AI).
- "My students should consider ethical issues from different perspectives of stakeholders, including developers, policy makers and users. They should not only explore ethical issues from different perspectives, but also develop principles for the ethical design and deployment of AI-based technologies" (AI teacher 10, impact of AI).
- "In my school, the test and examination of computer literacy assessed student knowledge of AI" (AI teacher 2, assessment of AI).

This finding agrees with most technical, skill-based and training programs that draw up lists of competences [16] and studies on teachers' concerns about assessment in new STEM curricula [25]. This could be because AI is considered to be one of subjects in the technology discipline; therefore, the teachers expected the AI curriculum would be assimilated into another existing technology subject.

Moreover, the analysis further showed that, in the content planning, there were many different technical, inconsistent and abstract terminologies, such as big data, cloud computing and machine learning. These unfamiliar and abstract terminologies would be detrimental for students during learning. In addition, different terminologies may be used to describe the same concept, which is at times confusing. These are reflected in following excerpts.

- "My students found the technical aspect too difficult and unfamiliar to understand" (AI teacher 1).
- "I need to spend a lot of time and effort to teach my student the AI terms" (AI teacher 3).
- "Is there any difference between deep learning and machine learning for school students?" (AI teacher 4).
- "I found many terms so abstract in AI and needed to suggest new ways to explain them to my students" (AI teacher 10).

The inconsistent and abstract terms complicated student learning by creating an extra and unnecessary cognitive load [26,27]. The teachers further suggested using consistent and familiar languages to facilitate the communications between teachers and students, and using graphical representations to present and explain abstract terminologies, knowledge and concepts. These could reduce confusion and provide directions to build teaching resources and also allow students to gain the foundation knowledge in the curriculum (see the following excerpts).

- "I used diagrams to explain what machine learning is" (AI teacher 2).
- "I communicated with my students using the term IPO (input–process–output—common terminologies)" (AI teacher 9).
- "I found many terms so abstract in AI and needed to suggest new ways to explain them to my students" (AI teacher 10).

Theme 2.2: Process and Praxis as the Pathway Towards Meaningful Learning

This subtheme indicates the process and praxis that form the core activities for students' sense-making of the content and product [7,15] consisted of experiential connection, student-centered learning and fear alleviation. The analysis revealed how important the experiential connection was in AI teaching. All the AI teachers highlighted that examples of real-world applications used in classrooms must be relevant to students in a local context, which connects the subject knowledge to the student's own experience (see the following excerpts).

- "I used KKBox (local and teenagers) instead of Spotify (global or adult) as an example to teach" (AI teacher 7).
- "Smart light (local and current issues) is the topic I used for my student inquiry task" (AI teacher 8).
- "My students experienced AI using their own experience and body movements" (AI teacher 9).
- "I used Hong Kong examples to teach AI" (AI teacher 10).

The analysis showed that all the AI teachers highlighted that the topic was similar to the other technology subjects that focus on technological capability, understanding and awareness. Therefore, after students gain foundation knowledge, they would use student-centered learning approaches such as design thinking, self-directed learning (setting learning goals) and project-based learning to teach the units.

- "I first identified the appropriate content and developed the slides. . . . Then I worked with my students to set their project title" (AI teacher 2).
- "My students used design thinking to develop an AI app to serve people. They set their learning goals (self-directed learning) for their learning" (AI teacher 5).
- "They created a smart (AI) home model proposal for their parents" (AI teacher 6).
- "Similar to coding, I planned to use project-based learning" (AI teacher 10).

The analysis further indicated that the AI teachers emphasized students' negative perceptions in the process and praxis. They expressed that the students had more negative perceptions about AI and tended to neglect the potential social good opened up by AI technologies. This is supported by the studies that generally find that peoples' perceived worries of loss of control of AI, ethical concerns about AI and the negative impact of AI on work have grown in recent years [18,28].

- "My students fear AI technologies" (AI teacher 2).
- "My students found AI scary (the end of the world), I use positive attitude to talk about AI now" (AI teacher 5).
- "My students had negative attitudes toward AI technologies" (AI teacher 9).

Theme 2.4: Renewal for Betterment

In this subtheme, there is a need for planning the sustained and iterative design of the curriculum; flexibility for revisable modules is seen as the design approach for AI curricula. As our analysis revealed, all the AI teachers were conscious about the need to renew curricula to update them with latest AI knowledge and education trends, and they reported that it was necessary to revise their teaching materials and improve pedagogy in cycles. The majority suggested that using a revisable module approach would offer a high level of flexibility in revising the curriculum. In this approach, all the teaching units should have no prerequisite knowledge and be independent (see the following excerpts).

- "I need to improve my teaching skills and update the content in cycles" (AI teacher 1).
- "There is no way that I will not revise the teaching materials. I have a lot to improve" (AI teacher 3).
- "We need to update the content as AI technologies are change rapidly" (AI teacher 4).
- "The teaching units must be explicitly designed for a specific goal (module) and be independent" (AI teacher 5).
- "Module-based curricula should be adopted. Easier to choose the unit for teaching and revising" (AI teacher 7).

*5.2. Theoretical Contributions*

5.2.1. Self-determination Theory

Our findings contribute to SDT by advancing support for the need for relatedness. As we previously discussed, studies on applying SDT to education have focused on the importance of autonomy and competency support for promoting intrinsic motivations [11]. Relatedness received less explicit focus in the literature, often being discussed in terms of self-oriented purpose [14]. In this self-oriented purpose, an individual actively considers their benefit as the first priority and seeks a sense of belonging to a specific group or community. For example, an individual prefers to join a group in which they can see where the benefits are.

In this study, SDT is viewed in the context of perceived needs for reform. The perceived needs are driven by the teachers' core values, including their students' well-being and future growth (see Theme 1.1). In other words, SDT is grounded in a sociocultural expectancy that has been internalized by the teachers. The teachers were motivated to plan AI teaching because they believed that AI knowledge is one of the competencies their students need for their future. The teachers took other (students') perspectives into consideration, connected with the school community and took the initiative to share their abiding missions, values and behaviors [12]. This could be due to the vocational call of duty and expectations from the society in which the teacher resides. Therefore, the need for relatedness could be supported by considering not only their own benefits but also other benefits in the community.

### 5.2.2. Curriculum Learning Approaches

As we discussed earlier, the process and praxis planning approaches were mainly adopted in established subjects, such as language and science; the content and product approaches were mainly adopted in technical subjects like physical education.

In this study, given that AI is an emerging discipline, the teachers seem to employ a mixed strategy and they are prepared to go through the design in an iterative manner. The curriculum would require multiple cycles of conceptualizing, deciding what to teach, how to teach, how to assess and consequently how to revise. It challenges current teacher education assumptions about the content knowledge, which are more likely known, stable knowledge, such as the basic knowledge of language and science [7,15].

Accordingly, we advocate that the curriculum planning approach needs to be considered from a comprehensive perspective for any emerging subjects. In this approach, the content input and product output delimit the problem space of the curriculum design, while the process and praxis form the core activities for students' sense-making of the content and product (Themes 2.1 and 2.2). From the teachers' perspectives, teacher-centric and student-centric curriculum approaches are both necessary and complimentary perspectives that enrich the curriculum design processes (Themes 2.1–2.3). The contextual factors, such as tools, resource and time availability and leadership support, on the other hand, are further necessary design foci to ensure that the lesson ideas can actually be implemented (Theme 1.3). On a personal level, the teachers' conviction for preparing students for their future drives the responsive and iterative curriculum efforts (Themes 1.1 and 2.3). In such a process, curriculum design processes engender teachers' growth of competence, which is cascaded down towards students' growth (Theme 1.2). In this sense, the curriculum creation efforts are anchored in design thinking and the vocational call of duty (Theme 1.1). Genuine curriculum creation thus encompasses and pays attention to all four forms of curriculum design approach, which are coordinated by the teacher's self-determination to be an ethical designer and orchestrator of students' learning experiences.

This curriculum planning approach not only guides school leaders and teachers on how to plan AI curricula, but also contributes to planning curricula for other emerging disciplines, like environmental science, and to other curriculum innovations such as interdisciplinary, transdisciplinary and phenomena-based programs.

### 5.3. Practical Recommendations

In this paper, we propose a five-stage curriculum development cycle for teachers, school partners and curriculum officers (see Figure 2). The five stages are given below.

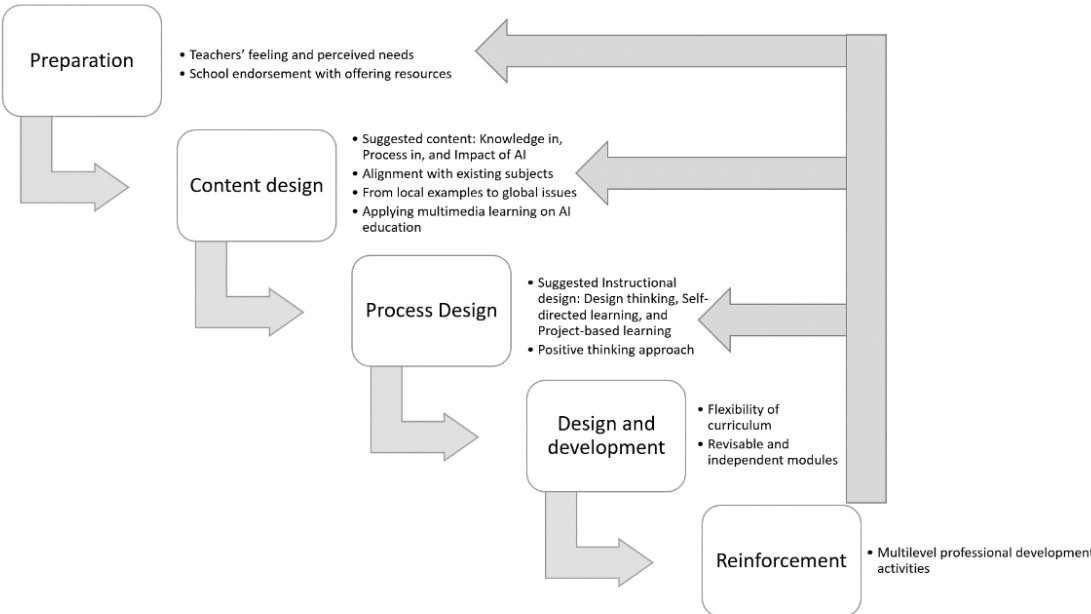

**Figure 2.** Our proposed development cycle for artificial intelligence (AI) curricula.

(1) Preparation: this stage is the most critical and focuses on the preparation of teachers for curriculum planning. How teachers feel and perceive AI teaching is more important than how well they know the AI content knowledge [9,10]. Therefore, schools should take immediate actions rather than make prolonged promises [10,25], and technology teachers prefer to receive direct, rather than indirect, support [9,25]. Schools should actively endorse the new initiative of AI teaching by offering different forms of support, such as logistical, technical and financial support, before it is requested by the teachers (Theme 1.3). The first relevant teacher training programs should build a sense of relatedness by focusing on helping teachers to understand the importance of AI K-12 education to their students and that AI education in schools is a global trend (see relatedness in SDT). This should be followed by substantial knowledge input to boost the teachers' efficacy in designing and teaching the AI curriculum. If possible, some autonomy about when and how the teachers should engage in AI training should be allowed.

(2) Content and product design: this stage is to identify content knowledge and assessment criteria, and methods for the curriculum. Aligning the AI curriculum with the assessment and objectives of existing technology or AI-related subjects is essential so that the innovation has clear and consistent guidelines with which the teachers are familiar. Moreover, to effectively transmit the abstract content knowledge, Mayers' multimedia learning principles, which facilitate generative, essential and extraneous processing, should be applied to develop teaching materials like slides, infographics and videos [26,27,29–32]. This will not merely transmit the content knowledge to students, but foster communication among teachers, students and content [31,32]. Finally, we suggest that the curriculum should adopt an approach of moving from local explanations to global understanding, making connections between the subject and the students' life. The teachers should use local problems as examples and then further extend to global issues; their students could have better understanding of the societal and personal impact of AI by combining many high-quality local explanations that allow representing global understanding.

(3) Process and praxis design: this stage is to design learning activities that encourage student development by addressing students' needs and interests, and alleviating AI fear through the design of socially meaningful group-based projects. To construct student-centered learning environments, we recommend that teachers should approach the subject area of AI from a design-thinking perspective with positive thinking that gains an empathic understanding of the people to design solutions of

the problems [17,33]. This approach supports different types of students to better understand the complex world of AI by ensuring the participation of more students [34]. Moreover, we recommended self-directed learning for hands-on software engineering tasks. This learning requires students to take ownership of their learning and involves interactions with the teacher and with other students.

(4) Design and development: in this stage the teachers design and develop materials for the curriculum based on the plan developed in the previous two stages. The teachers should apply a revisable and independent module in designing the curriculum (Theme 2.3), one which maximizes flexibility for school teachers not only to revise and teach the content based on their school environments and students' interests and competencies, but also to improve the materials and pedagogy. We believe that many variations may be needed around some key AI concepts to fit the diverse and dynamic needs of the students. In this approach, the teachers can choose the modules that they want to focus on for teaching or redeveloping.

(5) Reinforcement: the teachers should be encouraged and supported to revise the curriculum for development renewal. Multilevel professional development networks should be adopted to sustain curriculum development (Themes 1.2 and 1.3). This network exists both within and across school networks, and it includes joint-school professional networks, school leadership networks and professional-school networks [35]. Networking outside the school does not limit itself to working with other schools. There are, in the wider community, organizations that are competent in the field of AI, for example, AI professors and AI application developers who can support teacher professional development. These three-tier networks are interwoven together so as to create and nurture the capacity for sustaining the AI curriculum as well as to foster school organizational change.

## 6. Limitations and Future Directions

There are four limitations in this study. First, this study investigated AI perception among experienced and inexperienced teachers; hence, we need more work to understand how best to support teachers as they attempt to plan and design AI teaching and integrate AI technologies into their practice. An examination of the confidence levels of these two groups of teachers within their classrooms would yield useful data for future professional development.

The second limitation is that the study did not evaluate the effectiveness of the AI school curriculum. Research into effective curricula needs to be conducted. Effective ways to enhance students' AI identity and interest would yield more effective AI learning [36]. Students with stronger identity and interest are more likely to have greater persistence, which will be reflected in how successfully and for how long they pursue AI studies and careers [36]. Therefore, we suggest that more studies should be conducted on what content knowledge should be included and which instructional design should be adopted for enhancing AI identity and interest.

Thirdly, while this study proposes a new curriculum development cycle to support the promotion of AI K-12 education, more studies are needed to validate, enrich and refine the cycle. We suggest that this study could also be extended by additional studies on other emerging subjects and curriculum innovations.

Finally, given the richness of lived experience, this study portrayed two important themes based on the data we obtained. For the participants we interviewed, there are indications of theoretical saturation, but the research was conducted in a case setting. Future research may explicate more nuanced understandings about teachers' experiences in designing curricula for AI, especially from other cultural settings.

**Author Contributions:** Conceptualization, methodology, validation, formal analysis, writing—original draft, writing—review and editing, supervision, project administration: T.K.F.C.; conceptualization, writing—review and editing: C.-S.C. All authors have read and agreed to the published version of the manuscript.

**Funding:** This research received no external funding.

**Acknowledgments:** We thank Kobe Tsang and Rene Ng for helping to code the data in the analysis process.

**Conflicts of Interest:** The authors declare no conflict of interest.

**Appendix A**

Interview protocol:

1.  How to prepare new teachers to design and teach AI curriculum.

    *   What do you need to design and teach new AI curricula?
    *   What factors do motivate you to prepare the AI curriculum planning?
    *   Do you think you are capable of doing it? Why?
    *   Do you want to work with school leaders and other school teachers?
    *   What other support do you need or expect?

2.  How to plan and develop the AI curriculum and its content.

    *   How did/would you start the AI curriculum planning and teaching?
    *   How do you design the content and assessment?
    *   Which topics do you think are the most important for your students to know?
    *   Can you show me your work and explain?

3.  Logistical issues within a school environment.

    *   How did the school support you/what support did you expect from schools?
    *   Are there any logistical issues you have or expect for your curriculum planning and teaching?
    *   Do you need financial support when designing and teaching the curriculum?

4.  Teaching strategies and learning design.

    *   How did/will you teach the curriculum?
    *   What instructional approaches did you use in teaching AI curricula?
    *   What are the best learning approaches for students?

5.  How to refine the curriculum in an iterative manner.

    *   How did/would you improve the curriculum?
    *   Do you expect some help from outside?
    *   What are the characteristics of the curriculum?

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
