# Peer review of "Sustainable Curriculum Planning for Artificial Intelligence Education: A Self-Determination Theory Perspective"

_sustainability, doi:10.3390/su12145568_

Round 1

Reviewer 1 Report

Please check a few lines for clarity:

38 "teacher's choice of..."

270 how to refine...

278 deductive

442 spacing

504

566 "mixed strategy..."

604 "before requested by teachers...

651 study, and ?three?

Reviewer 2 Report

Review for paper titled “Sustainable Curriculum Planning for Artificial Intelligence Education: A Self-determination Theory Perspective”.

This paper presents a qualitative study where semi-structured interviews were conducted with practicing teachers to elicit their views on AI curriculum integration in K-12 education.

First, thank you for the opportunity to review this paper.

The authors present a very well-constructed literature review and adopted an appropriate method to address motivated research questions. There are parts of the article where I think the authors need to better justify some decisions they make, and the paper needs to be reviewed for minor spelling and grammatical errors, but overall I think the article makes a nice contribution to the pertinent literature.

I have attached a series of comments which I hope are beneficial to the authors.

Major considerations:

  1. The authors grounded their work in self-determination theory. This is appropriate, but the authors should include justification as to why this theory was selected.
  2. 24 participants were included. It is clear how they were sampled. I do not think the justification that 24 participants is enough for thematic codes to be created is sufficient though. Rules of thumb or heuristics such as this are not good for research. Was, for example, data saturation considered? Findings from thematic analysis tend to produce 3 themes regularly (in this case 3 sub-themes). This is a peculiar phenomenon considering the complexity of the lived experience of people which is inherent within the interpretivist paradigm. How do readers know the derived themes are a comprehensive set of themes from which to reflect on AI curriculum?
  3. I appreciate that the interviews were semi-structured, but I think from a transparency perspective it would be beneficial is the authors included the base interview protocol (questions asked) as an appendix. This would allow other researchers to adopt the protocol for their own studies, and for people to judge the scope of the initial questions.
  4. I was happy to see the authors did not report agreement between coders and did adopt a qualitative lens when analyzing their data. However, all qualitative analysis is influenced at least partially by the epistemological stance of the coders. I think details of the personal views of the coders with respect to AI in the curriculum should be included as well as details of relevant expertise.
  5. From the method section, there were more team members than there are authors. This is not to suggest the author list is not an accurate reflection of the authors, but are there people who deserve to be acknowledged for their contributions to the work? This is also important with respect to who was involved in analyzing the data (see comment 4 above).

Minor considerations

  1. In the abstract I would add (n = 24) to lines 19-20 to clarify is it 12 teachers from each group. When I first read it I misunderstood and thought it was 12 teachers in total.
  2. On line 206, the sentence “They are some important…” should be “There are some important…” I think.
  3. On line 253, the sentence “Therefore, we used a purposeful sampling…” should be “Therefore, we used purposeful sampling…” I think the “a” should be removed.
  4. On line 258, the sentence “This results in 18 male…” should be “This resulted in 18 male…”
  5. On line 261 the word “right” should be “rights”
  6. On line 278 the word “Deductive” should not have a capitalized “D”
  7. On line 664, the word “Final” should be “Finally”
  8. I have listed some spelling or grammatical errors here, but the paper does need to be reviewed thoroughly for spelling and grammar.
  9. This may be my computer causing a problem, but I am highlighting it just for the authors to check. Some of the sub headings appear to have the capital letters in a different font. For example, on line 250, for the section heading “4.1 Participants and Data Collection”, the capital letters D and C appear to be bold text and a different font. Again, this could just be on my laptop though.
